# TSSPREDATOR-WEB: A web-application for transcription start site prediction and exploration

Mathias Witte Paz [1]*, Alexander Herbig [2], Kay Nieselt [1]*

**1** Institute for Bioinformatics and Medical Informatics, University of Tübingen, Tübingen, Germany,
**2** Department of Archaeogenetics, Max Planck Institute for Evolutionary Anthropology, Leipzig, Germany

\* mathias-alexander.witte-paz@uni-tuebingen.de (MWP); kay.nieselt@uni-tuebingen.de (KN)

## Abstract

### Background

With the rapid development of high-throughput RNA-seq technologies, the transcriptome of prokaryotes can now be studied in unprecedented detail. Transcription start site (TSS) identification provides critical insights into transcriptional regulation. Still, current command-line tools for the prediction of TSS remain challenging with respect to their usability and lack of integrated exploration features.

### Results

We introduce TSSPREDATOR-WEB, an interactive web application that enhances the usability of the established yet unpublished tool, TSSPREDATOR. TSSPREDATOR-WEB facilitates TSS prediction from non-enriched and enriched RNA-seq data, classifies TSS relative to annotated genes, and allows users to explore results through dynamic visualizations and interactive tables. For the visualizations, we provide an UpSet plot summarizing the TSS distribution across experiments or classes, and a genome viewer that integrates transcriptomic and genomic data, which contextualizes the insights of the TSS predictions. To illustrate the usage of TSSPREDATOR-WEB, we provide a use case with Cappable-seq data from *Escherichia coli*. TSSPREDATOR-WEB is available on the TueVis visualization web-server at https://tsspredator-tuevis.cs.uni-tuebingen.de/.

### Conclusions

By combining user-friendly accessibility with interactive data exploration, TSSPREDATOR-WEB significantly facilitates genome-wide TSS analysis and interpretation in prokaryotes, empowering a broader range of researchers to generate biological insights from transcriptomic data.

**Data availability statement:** The raw data used for the provided use case can be found under the GEO accession number GSE215300. The processed data to reproduce the provided use case can be accessed from the main page of TSSPREDATOR-WEB https://tsspredator-tuevis. cs.uni-tuebingen.de/. If furtherly needed, the processed data can be uploaded in a public repository.

**Funding:** MWP and KN are supported by infrastructural funding from the Cluster of Excellence EXC 2124 'Controlling Microbes to Fight Infections' [project ID 390838134] from the DFG (Deutsche Forschungsgemeinschaft, German Research Foundation). We acknowledge support from the Open Access Publication Fund of the University of Tübingen. There was no additional external funding received for this study.

**Competing interests:** The authors have declared that no competing interests exist.

# 1 Introduction

Due to the development of high-throughput RNA-seq technologies, the transcriptome of a prokaryotic organism can now be studied in unprecedented detail. Far beyond the quantification of known transcripts, the precision of the data allows for the detection of novel transcripts or the definition of transcript starting boundaries [1,2].

To clearly define the base-specific starting boundary of a transcript, researchers require the identification of transcription start sites (TSS). Their identification plays a crucial role in understanding prokaryotic transcriptional regulation, as they define the exact positions where the RNA polymerase initiates transcription as well as the location of the promoter region. Promoter regions contain essential regulatory elements such as binding sites for various sigma and transcription factors [3,4]. This facilitates the generation of insights on how gene expression is regulated under varying environmental conditions [5,6]. Beyond this, the base-specific TSS identification also defines untranslated regions (UTRs), which contain riboswitches and RNA thermometers [7] and helps to identify other regulatory instances in bacteria, such as non-coding RNAs (ncRNAs) or antisense RNAs [8]. Despite the critical role of TSS identification, accurately predicting these sites remains challenging. Most genome annotations only provide predicted translation start sites and coding regions. Moreover, relying solely on standard RNA-seq data for a base-exact genome-wide TSS prediction does not produce comprehensive results, as this protocol does not distinguish unprocessed, so-called *primary* transcripts from processed transcripts (i.e., transcripts with a degraded 5'-end). Reads originating from processed RNA transcripts will shift the coverage profile towards the 3'-direction and thus bias the exact TSS prediction [1,9]. To overcome these challenges, special library preparation protocols have been developed for the determination of genome-wide TSS maps.

In 2010, Sharma *et al.* presented *differential RNA-seq* (dRNA-seq), an experimental approach to enrich for reads originating from the 5' ends of primary transcripts (i.e., transcripts containing a $5'$-triphosphate instead of a monophosphate) in prokaryotes [1]. This is achieved by treating the cDNA with a terminator exonuclease (TEX), which specifically degrades processed RNAs with a 5'-monophosphate. A more recent method for enrichment is Cappable-seq [2]. Instead of degrading processed RNAs, a vaccinia capping enzyme (VCE) is used to positively enrich primary RNAs with a 5'-triphosphate end. A more general approach is tagRNA-seq [10], which attaches specific *tags* to primary and processed RNA molecules. Besides the difference in methodology, all enriched libraries can be sequenced to produce enriched expression profiles, thereby increasing the sensitivity and specificity of TSS annotation strategies, making genome-wide TSS identification possible. Such a comprehensive view of transcriptional activity enables the identification of not only local regulatory elements but also global regulatory patterns throughout the organism, such as unannotated ncRNAs throughout a genome [11]. With the genome-wide TSS mapping, a high-resolution view of gene regulation is provided by identifying differences in promoter usage, identification of the length of 5' untranslated regions (5'-UTRs), and the presence of novel transcripts across species, going beyond what the typical RNA-seq experiments offer, in terms of comparative transcriptomics [12].

However, as the scale of the data generated by these approaches grows, manual curation becomes impractical, necessitating the development of fully automated computational methods to process and analyze TSS data efficiently.

Various tools have been developed to automate the prediction of TSS from enriched RNA sequencing libraries. Here, we focus on one of the first methods developed for TSS prediction, TSSPREDATOR [12]. Other tools, such as TSSAR [13] and TSSer [14], have been developed to identify TSS from standard and dRNA-seq data using statistical models based on the Poisson and Skellam distributions or Bayesian approximation based on the binomial distribution, respectively, to identify significantly enriched primary transcripts. A newer approach, ToNER [15], has been developed for the analysis of Cappable-seq data. Similarly to the previous methods, it relies on the computation of an enrichment score for each position and the identification of those that are statistically significant. Lastly, a support vector machine learning-based approach has also been explored to improve predictive accuracy [16]. Differently from all previous methods, TSSPREDATOR follows a heuristic approach by mimicking the manual TSS annotation process originally described by Sharma *et al.* [1]. TSSPREDATOR identifies peaks in expression across the genome by comparing the values to thresholds set by the user. Afterwards, enrichment scores are computed by comparing both RNA-seq libraries (enriched and standard) to identify transcription start sites, without relying on a statistical approach. The latest version of TSSPREDATOR now includes support for three prominent experimental protocols (dRNA-seq [1], Cappable-seq [2], and tagRNA-seq [10]), alongside new features, such as the analysis of multi-contig assemblies. While neither TSSPREDATOR nor its further developments have been released as a stand-alone publication, the tool has proven effective in various studies. For instance, it has been used to compare the transcriptome of one organism under different conditions (e.g., [6,17]), as well as cross-strain analyses to identify conserved and divergent transcriptional characteristics (e.g., [12,18]).

Still, given the complexity and volume of genome-wide TSS data, TSSPREDATOR as well as the other command-line tools mentioned above, often fail to provide the efficient data exploration needed for meaningful interpretation and insight generation. Interactive web applications address this challenge by simplifying both the use of the tool and the exploration of results through user-friendly interfaces. By including dynamic and interactive representations of TSS distributions, promoter sequences, and expression patterns, such platforms enable users to quickly inspect genomic regions, compare transcriptional landscapes, and identify patterns that might otherwise remain hidden. Features such as summary visualizations, interactive genome browsers, expression data overlays, and customizable filtering options enhance accessibility and facilitate hypothesis generation, making genome-wide TSS analysis more accessible, efficient, and informative.

To achieve this, we introduce TSSPREDATOR-WEB, a web application designed to predict and explore TSS identified by TSSPREDATOR. The web interface facilitates the interaction with TSSPREDATOR, for example, by reducing the hurdles of dependencies installation and by providing enhanced interactions for data upload and data sharing. Moreover, TSSPREDATOR-WEB enhances the exploration of results by offering interactive visualizations, providing an overview of genome-wide TSS maps, as well as detailed views on the TSS predictions, thus facilitating deeper insights into transcriptional regulation.

## 2 Methods

### 2.1 Comparative detection of TSS

Before introducing TSSPREDATOR-WEB, we introduce the underlying algorithm of TSSPREDATOR for TSS detection. TSS-PREDATOR is able to analyze different sequencing protocols for TSS detection in prokaryotes, as long as they produce a control and an enriched library. So far, it has been tested with data provided by all established protocols mentioned above: dRNAseq [1], Cappable-seq [2] and tagRNA-seq [10]. The sequencing results need to be processed to produce coverage profiles of the complete reads for both libraries, either in *wiggle* or in *bedGraph* format, using mapping workflows such as `READemption` [19]. TSSPREDATOR then identifies the TSS by comparing both profiles and expecting an increase in expression for the TSS in the enriched library versus its non-enriched counterpart, independently for each genomic strand (forward and reverse). For this, TSSPREDATOR expects the expression data pairs (enriched and non-enriched) to be

normalized beforehand. For example, one possibility provided by `READemption` is to normalize the coverage by the total number of aligned reads and then multiply each position by the lowest number of aligned reads of all considered libraries (*coverage-tnoar_min_normalized*) or by a million (*coverage-tnoar_mil_normalized*). TSSPREDATOR then conducts a further normalization between libraries (see S1 Algorithm). For each enriched library, by default, the 90th percentile ($Q_{PL}$) is computed and used as a factor for percentile normalization of both libraries—enriched and its corresponding non-enriched counterpart. To recover the original data range, the minimal percentile value across samples ($Q_{min}$) is multiplied by the normalized values. Moreover, to normalize for different enrichment factors across replicates and sets, the median enrichment factor (i.e., enriched value divided by non-enriched library) is computed by default for all library pairs (enriched and non-enriched). The largest enrichment factor $EF_{max}$ is then used to normalize all non-enriched libraries. While not activated by default, TSSPREDATOR can export the normalized libraries for further downstream analyses after TSS prediction.

After normalization, for each position $i$ in each enriched expression graph, TSSPREDATOR extracts the expression height $e(i)$, and computes, with respect to the previous position $e(i-1)$, the height change $e(i) - e(i-1)$ and the factor of height change $\frac{e(i)}{e(i-1)}$ (Fig 1, S2 and S3 Algorithms). These values are compared to predefined thresholds. If a position $i$ exceeds the thresholds for step height $T_h$ and step factor $T_f$, it is considered a TSS candidate and classified as *detected*. Here, it is important to note that the step height threshold is relative to the 90th percentile value used for the *inter-library* normalization. If too many of the detected TSS are found close to each other within a window size $W$, they are reduced by selecting either only the first TSS or the one with the highest expression. This produces a set of putative TSS per strand and per replicate.

In case of more than one replicate per experiment (that is, an organism's genome or a tested condition), predicted TSS that are within a distance of 1 bp when comparing replicates are considered equal to allow a cross-replicate shift ($\Delta_r$). This default value can be changed by the user. If a TSS is labeled as *detected* in one replicate, the corresponding positions in the other replicates are re-evaluated by reducing the thresholds by predefined *reduction values* for the step height and step factor ($\rho_h$ and $\rho_f$, respectively). This increases the number of detected TSS across replicates. A TSS needs to be detected in a minimal number of replicates ($R_{min}$, default: 1) to be included in the next steps. However, this lower threshold can be increased for higher specificity. For all detected TSS at any position $j$, the enrichment factor $EF = \frac{e_{enr}(j)}{e_{non}(j)}$ (Fig 1, S2 Algorithm) is computed for all replicates with respect to the same position in the non-enriched library, and the maximal value across replicates is compared to the enrichment fact threshold $T_{EF}$. For example, if the selected threshold is 2, it means that the double expression value is expected in the enriched library compared to the control library for the

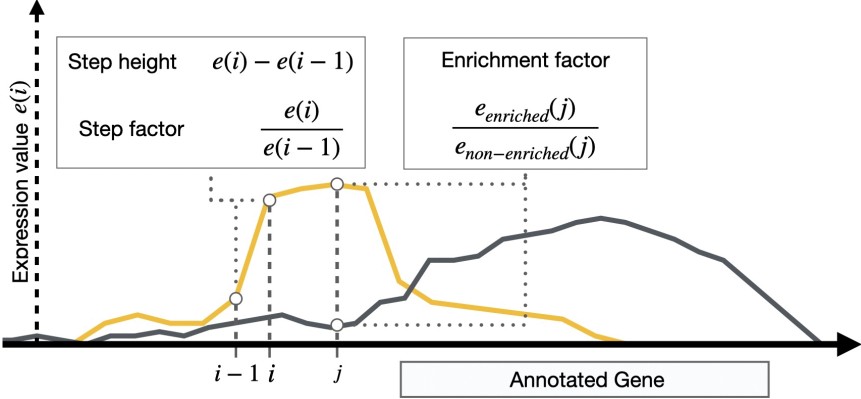

**Fig 1. Sketch of the computation of the three parameters *step height*, *step factor* and *enrichment factor* required for TSS prediction.** The yellow line shows the enriched profile ($e_{enr}$), with an increased expression value upstream of the annotated gene.

TSS to be labeled as enriched. Analogously to the cross-replicate shift, all TSS found within the cross-condition shift value (default: 1 bp), are considered equal. This returns one set of detected and enriched TSS per strand and experiment.

Lastly, the sets of enriched TSS per experiment are compared to each other. The comparison of TSS in a *multiple-condition* experiment is straightforward since the TSS are already in the same coordinate system. For a *multiple-strain* experiment, a common coordinate system is computed using the concept of the *SuperGenome* based on a whole-genome alignment [12], such as the `XMFA` file provided by `Mauve` [20]. With the alignment, TSS are compared to each other and clustered together if they are found in close chromosomal distance. This returns, per experiment, a set of detected and enriched TSS that can be further analyzed. All mentioned thresholds can be modified by the user to lower or increase the specificity and/or sensitivity of the prediction. As the prediction step is influenced by the chosen thresholds and parameters, five predefined parameter sets have been introduced to span a range of sensitivities and specificities, from *very sensitive* to *very specific* (S1 Fig). An overview of how the parameter sets influence the TSS-calling process for an experiment with one condition and one replicate is provided in S2 Fig.

Finally, all detected TSS are classified according to their locations relative to annotated genes. While other classification methods exist [21], here we classify TSS as defined in [1] (Fig 2). 5 different classes are defined: Primary and secondary TSS are found up to 300 nt upstream of a gene's annotated translation start, where the strongest or the first signal are classified as primaries, and the rest of signals as secondary TSS. Internal TSS are found within an annotated genes, and antisense TSS are found on the antisense strand within a distance of less than 150 nt to an annotated gene. Lastly, TSS that are not associated with either of the other four classes are called orphan TSS. Note that a TSS can also get more than one class assignment (e.g., it can be a primary TSS of a gene and at the same time an antisense TSS for a gene on the opposite strand). The resulting predictions are stored in a TSV-file called *MasterTable*, which summarizes all relevant information per TSS. This table describes in detail all predicted TSS positions, such as their enrichment factor, their class, the gene to which they might be associated, among others. This is reported for every experiment in the analysis, showing multiple lines per TSS and classification. Moreover, TSS<small>PREDATOR</small> provides GFF files for each experiment aligned into one coordinate system. This is especially useful when analyzing multiple strains of a bacterium.

### 2.2  Design process of TSS<small>PREDATOR</small>-W<small>EB</small>

The workflow for TSS prediction and their association with a gene is run independently for each replicate, condition, and strand, as described above. For each replicate, four input files are required—one for each strand (forward and reverse)

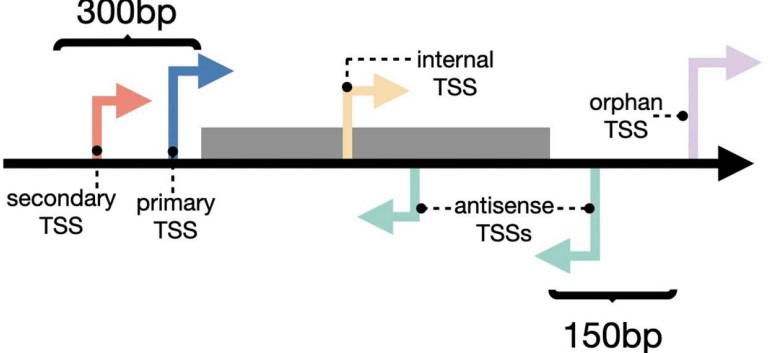

**Fig 2.  Classification of TSS based on the distance to annotated genes as defined by [1].** Primary and secondary TSS are located upstream of annotated genes, where secondary TSS show a lower enriched expression signal compared to the respective primary TSS. Internal TSS are located within the genes themselves, while antisense TSS are located on the antisense strand close to a gene (within 150 bp) or within the gene itself. Lastly, all other identified TSS are called orphan TSS.

and experimental type (non-enriched control and enriched). Since TSSᴘʀᴇᴅᴀᴛᴏʀ expects each file to be correctly categorized, every file must be uploaded separately.

To facilitate this interaction, the latest version of TSSᴘʀᴇᴅᴀᴛᴏʀ offers a `JAVA`-based GUI to assist users in allocating files and setting the required parameters for the TSS prediction (https://it.inf.uni-tuebingen.de/tsspredator, accessed on May 2025). From this GUI, the data processing can be initiated, and the predicted results are generated and saved in external files for downstream exploration.

While effective for TSS prediction, this process presents several usability challenges. First, since TSSᴘʀᴇᴅᴀᴛᴏʀ is implemented in `JAVA`, it requires users to install additional dependencies, limiting platform independence and posing difficulties for users with limited technical expertise. Second, the manual file allocation process becomes increasingly cumbersome, and possible error-prone, as the number of replicates and conditions grows. Finally, downstream analysis is not integrated into the tool, so that users must export the results to other platforms to gain a comprehensive overview or biological insight.

By reviewing several genome-wide TSS studies, we identified common strategies used to explore such data sets. For example, many studies provide overview visualizations with respect to the TSS distribution across their classes or analyzed conditions [6,16,22]. Additionally, these studies often integrate transcriptomic and genomic data, especially gene annotations and upstream regulatory regions, within one view for enhanced data exploration [1].

Based on the limitations of TSSᴘʀᴇᴅᴀᴛᴏʀ, and the evaluation of published TSS studies, we identified eight requirements to improve the TSS analysis workflow (see Fig 3). These requirements can be classified with respect to the step of the TSS prediction workflow they can improve: the input allocation (R1, R2), data processing (R3, R4) and result exploration (R5, R6, R7, R8).

Building on these requirements, we have defined a set of goals to be addressed by TSSᴘʀᴇᴅᴀᴛᴏʀ-Wᴇʙ:

**G1 Accessibility (R3):** Ensure platform independence by eliminating installation requirements, allowing users to easily access the tool.

**G2 Usability (R1, R4–R7):** Simplify the user experience by providing easy ways of uploading and exploring the data, as well as making long waiting times bearable.

**G3 Exploration (R5–R7):** Support data exploration through interactive visualizations that present TSS predictions across levels of detail.

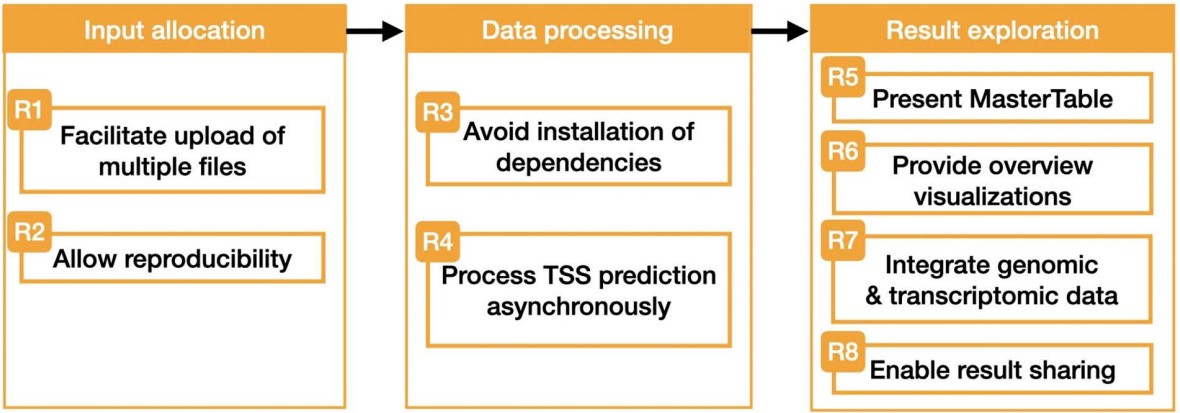

**Fig 3. Requirements identified for the improvement of the TSS prediction workflow.** Each of them will be tackled with the implementation of TSSᴘʀᴇᴅᴀᴛᴏʀ-Wᴇʙ.

**G4 Data Integration (R6, R7):** Facilitate exploratory analyses by combining genomic and transcriptomic information into an interactive view, providing genomic context of the results.

**G5 Reproducibility and Data Sharing (R2, R8):** Provide mechanisms to reproduce and share both the prediction results and exploration processes with others.

### 2.3 Back-end of TSSpredator-Web

To facilitate the accessibility for the TSS prediction workflow, TSSpredator-Web has been designed as a web-application. It is freely accessible via the TueVis visualization platform (https://tsspredator-tuevis.cs.uni-tuebingen.de). However, we also provide Docker images that facilitate deployment on any server, as well as allow local usage of the tool (https://github.com/Integrative-Transcriptomics/TSSpredator-Web/releases). The back-end was designed with `Flask` (v2.3.3) and runs a `JAVA`-compiled version of TSSpredator for the TSS prediction. For multiple simultaneous asynchronous requests, a worker manager based on *Celery* (v5.3.4) and `redis` (v6.2.0) has been integrated in the back-end. The results of the TSS prediction are preprocessed using `Python` scripts and `bedGraphToBigWig` [23] for the subsequent interactive exploration in the web interface.

### 2.4 Web-interface

The front end of TSSpredator-Web has been developed using `React` and is structured into three distinct pages, each corresponding to one step in the TSS prediction workflow. The first step, *input allocation*, offers an improved file upload experience. Users can upload all necessary files at once and conveniently organize them using a *drag & drop* interface. The predefined sets of parameters described previously can be selected from this interface or adapted by the user. Once all parameters have been set, a configuration file that encompasses all the chosen parameters can be saved for future use, ensuring future reproducibility. If any user is provided with such a configuration file, they only need to upload it along with a folder containing all needed files, for TSSpredator-Web to allocate the inputs automatically, enabling a faster start of the analysis.

Upon starting the TSS prediction, users are redirected to the status page, where information on the data processing is regularly updated. This allows an effortless monitoring of the analysis without needing to keep the browser window open, as well as facilitating the sharing of ongoing predictions. Additionally, if any result have already been computed with TSSpredator-Web, the users are able to upload a `ZIP`-file of these results. This skips the need to rerun the prediction and still enabling access to the downstream data exploration features of TSSpredator-Web.

### 2.5 Data exploration & integration

To support the goals of data exploration & integration, as well as the usability improvement, TSSpredator-Web completes the TSS prediction workflow by providing an interactive result analysis. For this, the *MasterTable* as computed by TSSpredator is presented as an interactive table, complemented by two visualization approaches: an UpSet plot [24] offering a high-level overview of the dataset, and a genomic viewer for contextualization of the TSS predictions.

The interactive *MasterTable* includes common features of exploration, such as searching, filtering, and sorting. Since TSS can be associated with more than one gene and therefore can be classified into more than one of the classes mentioned above, they correspond to one row in the *MasterTable* per classification. The TSS distribution among the different classes or between experiments is visually summarized by an UpSet plot [24]. By setting which variable should be used for the plot (either TSS classes or different experiments), the users can identify how many TSS positions have multiple TSS classes or how many of them occur in specific combinations of experiments. To get more information on these subsets of TSS, these subsets in the UpSet plot are interactive, such that users can select them, and only their corresponding rows are shown in the *MasterTable*.

   

For the exploration of TSS in a genomic context, a genome browser has been implemented. Based on the visualization grammar `Gosling` [25], the genome browser provides aggregated and detailed views of the data for each strand independently, following Shneiderman's mantra: *Overview first, zoom and filter, details on demand* [26]. The aggregated view (Fig 4, top) consists of multiple visualization components. The main component shows a stacked bar chart that bins the TSS according to their position in the genome and their assigned class. Depending on the zoom level, the view is aggregated with bins of 50kbp, 10kbp or 5kbp. A further track below the stacked bar charts indicates annotated genes by including gray rectangles at the corresponding position. Such an aggregated view is shown until a full window size resolution of 50,000 bp is reached. From this point on, the detailed view (Fig 4, bottom) is displayed, showing each individual TSS and the surrounding annotated genes. Within this view, the main track also includes the normalized expression values for the enriched and control libraries at each genomic position. To provide information on gene regulation, the 50 bp-long sequence upstream of the TSS is visualized on a third track. A tool-tip can provide more detailed information of each TSS, gene, and expression value.

The described tracks visualize the data separately for each strand, with a difference in the orientation of the plots and glyphs (Fig 5). This mimics the visualization method used in other genomic browsers, such as the *Integrated Genome Browser* (*IGB*) [27] or the *Integrative Genomics Viewer* (`IGV`) [28]. The *genome browser* of TSSPREDATOR-WEB provides two view arrangements to facilitate the exploration of the data within one experiment, and also the comparison across experiments (Fig 5). The *single view* mode groups the tracks of each experiment together (see Fig 5, top row), thus facilitating the exploration of single experiments, under different conditions, or single strains of one organism. Furthermore, the *aligned view* mode (see Fig 5, bottom row) groups the components vertically with respect to their strand, hence facilitating

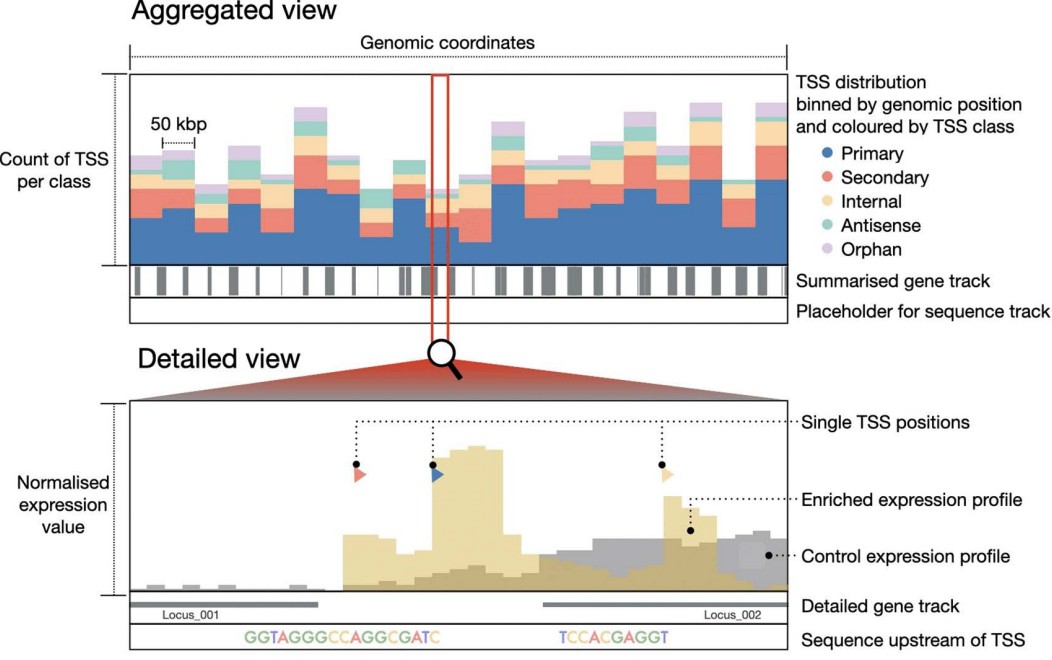

**Fig 4. Sketch of different views of the *genome browser* of TSSpredator-Web.** The top section (aggregated view) displays the aggregated stacked bar chart. Each bar shows the count of TSS of a specific class in the respective genomic region bin. Below, the gene track provides a hints for the location of annotated genes. The bottom section illustrates how the plot changes upon falling below the 50,000 bp threshold (detailed view). Individual TSS locations are represented by colored glyphs, and genes are shown in their entirety with their corresponding name or locus_tag. The visualization also includes expression data from both control (in gray) and enriched libraries (in yellow), along with a track that illustrates the upstream region of a TSS. For simplicity, only 10 bp of the upstream region are shown on this representation instead of the actual 50 bp.

**Fig 5. Visual representation of the two modes of the genome browser to show data of multiple experiments.** Each experiment consists of one component per strand, differing on the orientation of the data (for example, the reverse strand is flipped vertically). The single view mode (top section) groups the components based on the experiments for a direct exploration within each experiment. Differently, the aligned view (bottom section) groups the visualization components vertically with respect to the two strands. This allows an easier comparison across experiments. Regardless of the chosen view, a synced crossline is shown on hover. For simplicity, in this figure this line is only shown in the single view.

the comparison between conditions or strains. Regardless of the chosen arrangement, all views are synchronized, so that the same zoom level is shown in all instances. Moreover, a synced cross-line appears on hover to identify the position over all views. The *genome browser* can be freely used for exploration at any level of detail of the genomic coordinates. If users require specific TSS positions, these can be searched in the *MasterTable* and then directly visualized in the *genome browser*.

The complete predicted dataset as well as each single visualization can be downloaded from the interface. Moreover, as each prediction of TSSPREDATOR receives a unique URL, the TSS predictions can be easily shared and accessed up to seven days after the corresponding TSS prediction was run.

## 3 Use case

To provide an example of how TSSPREDATOR-WEB can be used to generate and analyze genome-wide TSS data, a dataset for *Escherichia coli* K-12 MG1655 published by Balkin *et al.* [29] [GEO Accession No. GSE215300] is used in the following section. In this study, the authors treated the *E. coli* strain with three different antibiotics (novobiocin, rifampicin and tetracycline). All three antibiotics have different modes of action. Novobiocin inhibits the DNA gyrase and, hence reduces

DNA replication [30]. Rifampicin blocks the bacterial RNA polymerase, reducing the RNA synthesis [31]. Lastly, tetracy-cline binds the 30S ribosomal subunit and inhibits protein synthesis [32]. For all three treatments and a control condition, the authors measured gene expression using three replicates per condition, following the Cappable-seq protocol [2] to generate 5′-enriched reads. Non-enriched VCE-capped reads generated using the *NEBNext Ultra™ II Directional RNA* prep kit are also available under the same GEO accession number. `READemption` [19] was used to align the reads and to compute the coverage plots in *wiggle* format. Enriched and non-enriched reads were aligned independently to the *E. coli* reference genome [NCBI Accession No. GCF_000005845.2], taking into account their different protocols for library preparations. The `READemption coverage` command provides different normalized *wiggle* files. To make both inde-pendent runs comparable, the `tnoar_mil_normalized` coverage plots were used for further analysis, consisting of 24 *wiggle* files (that is, 4 conditions × 3 replicates × 2 strands) per library protocol (non-enriched and enriched). These 48 wiggle files, together with the genome and annotation file obtained from NCBI, are the basis for the prediction of TSSPREDATOR-WEB.

Since one of the goals of TSSPREDATOR-WEB is to provide a user-friendly way to predict and explore genome-wide TSS maps, this starts already with the upload of the data. The 50 required files can be easily uploaded to TSSPREDATOR-WEB using the *drag & drop* functionality (S3 Fig). Here, the files were distributed among the four conditions and three repli-cates. To increase the confidence of the results, the predetermined *very specific* parameters were chosen for the TSS pre-diction step. For clustering after prediction, a cross-condition shift of 3 bp and a cross-replicate shift of 2 bp were allowed. Based on these parameters, TSSPREDATOR identified a total of 7, 194 genomic positions as enriched TSS.

The genome-wide TSS exploration process starts with an overview of the TSS across classes and analyzed conditions. Taking into account only the genomic position of a TSS, the UpSet plot shows that most TSS are classified as internal, directly followed by primary TSS (Fig 6A). However, this distribution does not account for how often a TSS occurs across conditions, meaning that a TSS can be enriched only in one condition. This can be verified by considering both, the genomic position and the condition in which the TSS occur for the UpSet plot (Fig 6B). The results show that primary TSS are, in fact, the predominant class in the results. A similar result can be seen by looking at the *genome browser*, where all TSS positions are aggregated by class and genomic position per condition (Fig 6C), where a predominance of primary TSS can also be observed.

To analyze the distribution of primary TSS even further, one can visualize the occurrence of this class across condi-tions. This can also be inspected via the UpSet plot (Fig 7A) and provides a glimpse into an interesting subset of TSS: 43 TSS surpass the threshold for detection only in the samples treated with any of the three antibiotics, but not in the control samples (highlighted in Fig 7A). These positions can be analyzed in more detail by interacting with the UpSet plot to filter the *MasterTable* for this specific subset. From here, the *MasterTable* can be sorted by step height (that is, the increase in the enriched library at position *i* compared to the previous position *i* − 1) to show the most prominent positions (an excerpt of the *MasterTable* shown in Table 1). Based on these results, users can search for further information, for example, by searching for more information about genes in known databases, such as the `EcoCyc` database [33] for *E. coli*. Some of the genes found in Table 1 were manually searched in `EcoCyc` to exemplify this exploration workflow. For example, the genes *ugd*, *arnB* and *ais*, are described as responsible for changes in membrane lipopolysaccharides (LPS), indicating a reaction against the hostile environments [34–36] as already shown for another antibiotic polymyxin [35]. In addition, the gene *dinI* indicates DNA damage, which can be caused by antibiotics, even though it is not part of their active mode of action [37]. In summary, some of the TSS with the highest step height and their associated genes reflect how *E. coli* reacts to the high stress caused by antibiotics.

Although providing an overview of the most prominent TSS can be helpful, combining this information with the tran-scriptomic layer provides even more insight. This can be achieved through the *genome browser*. Here, we inspect the most prominent TSS with respect to the step height: position 2, 099, 734, the primary TSS of gene *ugd* (locus_tag: b2028, Fig 7B). Here, it can be seen that enrichment libraries show the highest expression for the TSS under treatment with

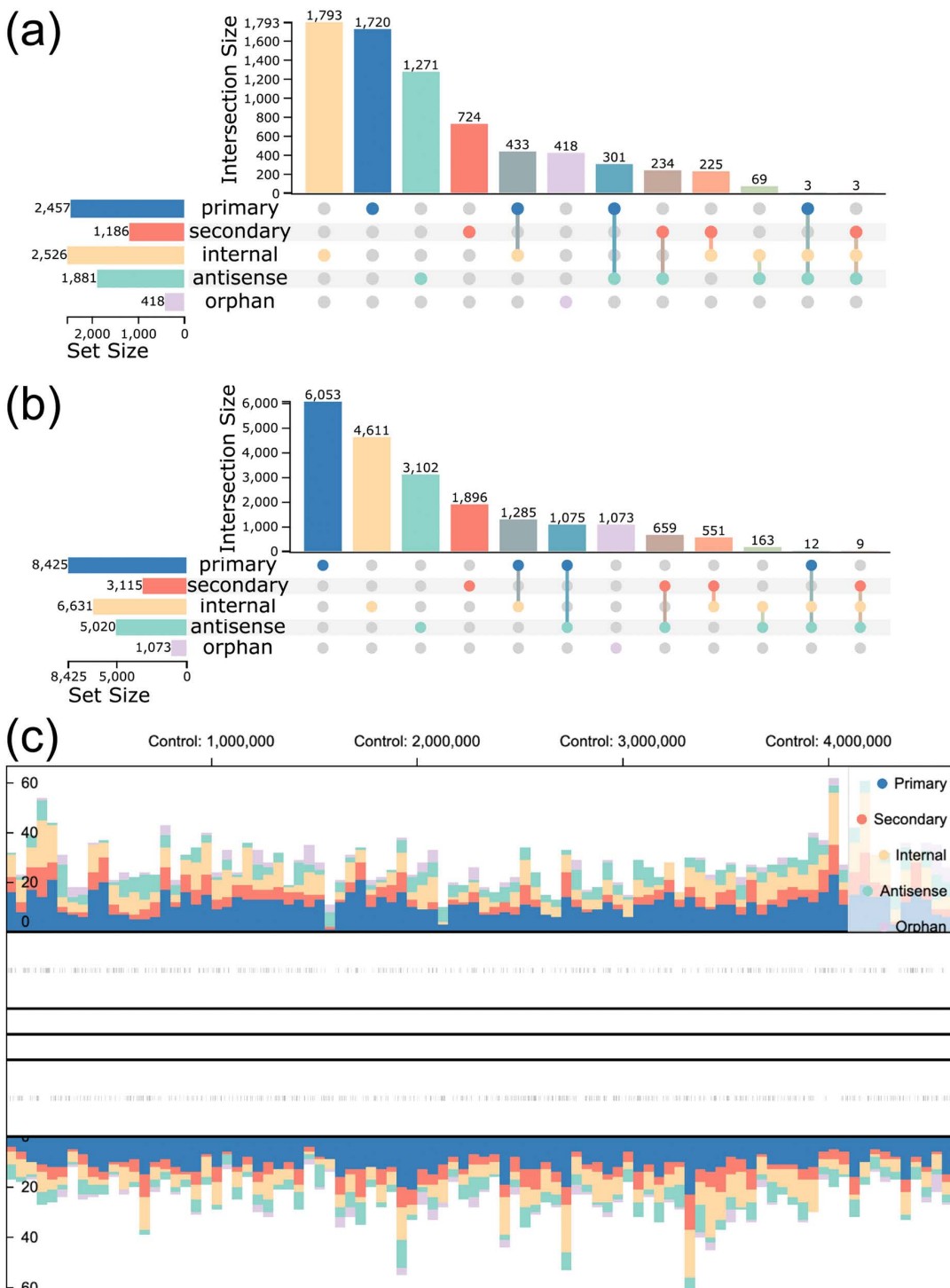

**Fig 6. Analysis of the overall distribution of TSS across conditions, classes and location in the genome for data collected for *E. coli* on four different conditions (one control and three treatment with antibiotics). (a)** UpSet plot showing the distribution of enriched TSS aggregated only by their location (i.e., position and strand). **(b)** UpSet plot showing the distribution of enriched TSS aggregated by their location and the condition they occur. For this UpSet plot, each TSS is counted for each condition separately. **(c)** Aggregated view of the *genome browser* showing the distribution of TSS colored by class and binned by their position in the genome for the control condition.

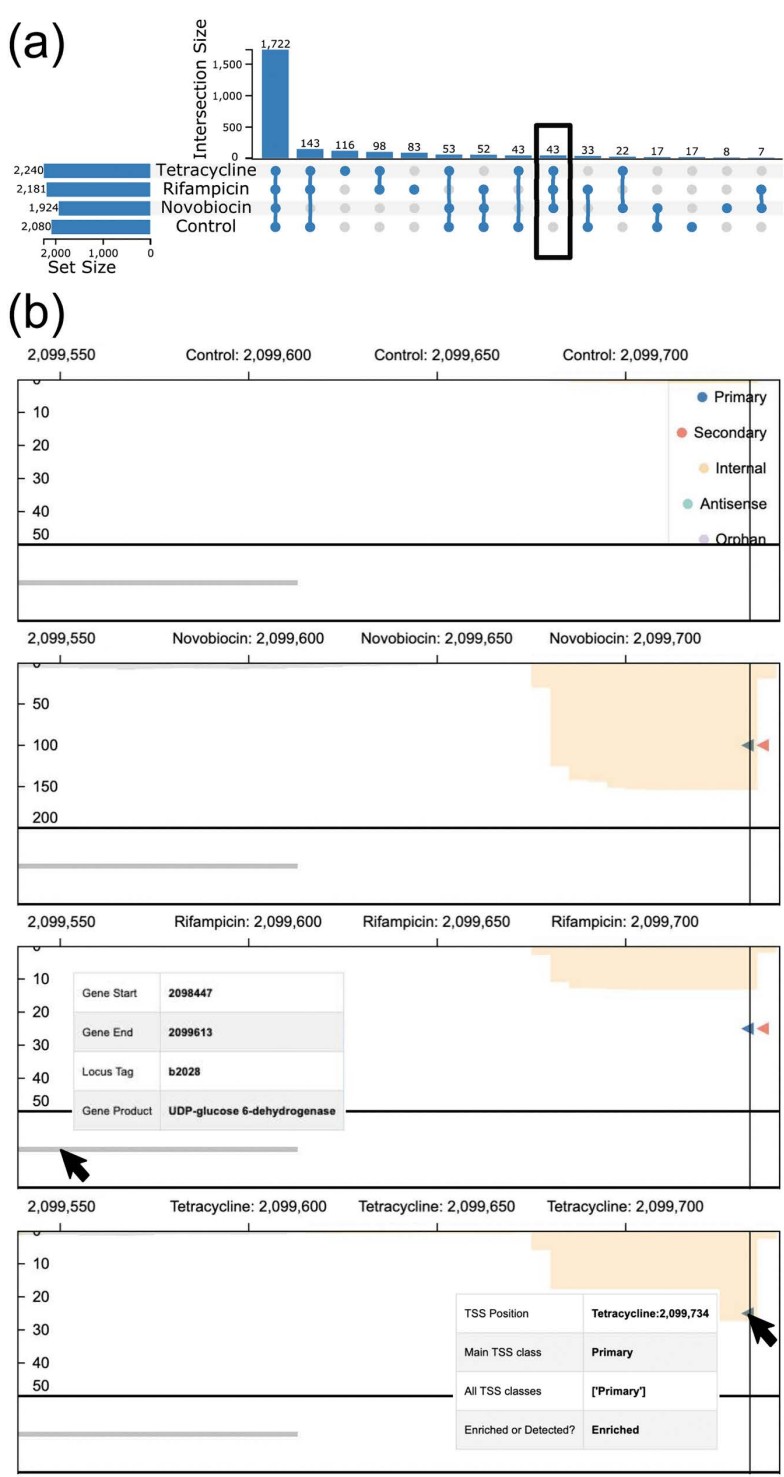

**Fig 7. Analysis of primary TSS for *E. coli* across conditions, especially those TSS occurring only under the treatment with each antibiotic. (a)** UpSet plot showing the distribution of primary enriched TSS across conditions, aggregated only by their location. The highlighted set refers to those TSS positions enriched only under the treatment with each antibiotic. **(b)** Aligned mode of the *genome browser* showing a primary TSS position on the reverse strand (shown via the direction of the glyphs and the bars of the expression profiles) occurring in all conditions with antibiotic treatment, but not in the control condition. The TSS is located upstream of the gene *udp* (locus_tag: b2028). The enriched libraries (orange bars) can be seen increased in all conditions. However, the treatment with *novobiocin* shows the highest expression value. For simplicity, the empty *genome track* has been removed from the figure.

**Table 1. Excerpt of the _MasterTable_ showing the top 10 enriched primary TSS enriched based on step height and their associated genes. These TSS are enriched in all antibiotic treatment conditions for _E. coli_.**

| Pos | Strand | stepHeight | Locus_tag | Gene Name | Product |
|---|---|---|---|---|---|
| 2099734 | – | 316.07 | b2028 | ugd | UDP–glucose 6-dehydrogenase |
| 712851 | + | 158.06 | b0687 | seqA | negative modulator of initiation of replication |
| 4275169 | – | 123.71 | b4060 | yjcB | uncharacterized protein YjcB |
| 3775568 | + | 75.8 | b3601 | mtlR | transcriptional repressor MtlR |
| 1056182 | – | 74.31 | b0993 | torS | sensor histidine kinase TorS |
| 3210566 | – | 74.27 | b3064 | tsaD | N(6)-L-threonylcarbamoyladenine synthase, TsaD subunit |
| 1121509 | – | 39.35 | b1061 | dinI | DNA damage-inducible protein I |
| 2365854 | + | 34.77 | b2253 | arnB | UDP-4-amino-4-deoxy-L-arabinose aminotransferase |
| 4392704 | + | 33.23 | b4670 | yjeV | uncharacterized protein YjeV |
| 2365723 | – | 28.99 | b2252 | ais | putative lipopolysaccharide core heptose(II)-phosphate phosphatase |

_novobiocin_, in comparison to the other two antibiotics. A recent study identified that among these three antibiotics, the membrane LPS modification triggered by _ugd_, among other genes, is most effective against _novobiocin_ [38]. A further step to analyze this region beyond TSSPREDATOR-WEB would be to extract the promoter and/or the UTR region of this gene to analyze putative regulatory elements in detail. Here, the UTR region of _ugd_ was manually extracted using the coordinates provided by TSSPREDATOR-WEB, and compared to the RFAM [39] database outside of the presented interface. Though no hit was identified, the secondary structure of the sequence was computed using RNAfold [40] and returned a stable secondary structure (see S4 Fig).

TSSPREDATOR-WEB also enables the analysis of another particularly interesting class of TSS: orphan TSS. These positions correspond to transcription start sites that cannot be associated with any annotated gene from the provided file, suggesting the presence of transcriptional activity outside known gene boundaries. Such signals may represent transcriptional units that were overlooked by standard gene annotation pipelines and become detectable only through genome-wide TSS mapping. To investigate such putatively overlooked genes, we analyzed the top 10 orphan TSS present under all conditions, ranked by their step height. A prominent region is close to the orphan TSS at position $2,904,461$ on the reverse strand (Fig 8A). Upon zooming in on this region in the _genome browser_, a noticeable increase in the enriched library can be observed, with expression levels increasing up to 350. Moreover, the upstream sequence contains a subsequence similar to the Pribnow box (TATAAA) at −9 nt upstream of the TSS (Fig 8B), suggesting a binding site for the RNA polymerase. When exploring the nearby regions, a second orphan TSS is identified on the forward strand at position $2,903,986$ (Fig 8C). This TSS also shows a clear Pribnow box at position −13 nt upstream of the TSS. Together, these two orphan TSS may represent previously unannotated transcriptional units. Their pronounced step heights and well-defined upstream promoter motifs make them strong candidates for further computational and experimental validation, with the potential to improve the genomic annotation of the organism. For example, one could analyze the downstream regions of the TSS to identify possibly overlooked open reading frames.

## 4 Discussion and conclusion

Genome-wide TSS maps provide important information for the analysis of the architecture of the prokaryotic transcriptome [6,29] or even recent studies on bacteriophages [41,42]. These studies facilitate the definition of the regulatory promoter region of genes and provide clear signals for the identification of unannotated genes [3,4,11]. Due to the large complexity of the underlying data, computational methods are required to analyze the data and produce insights. The currently available methods tackle the prediction of TSS with different underlying methodologies [13–16] and most commonly provide only a command-line tool. Yet, the usage via the command-line demands technical expertise, which should not be

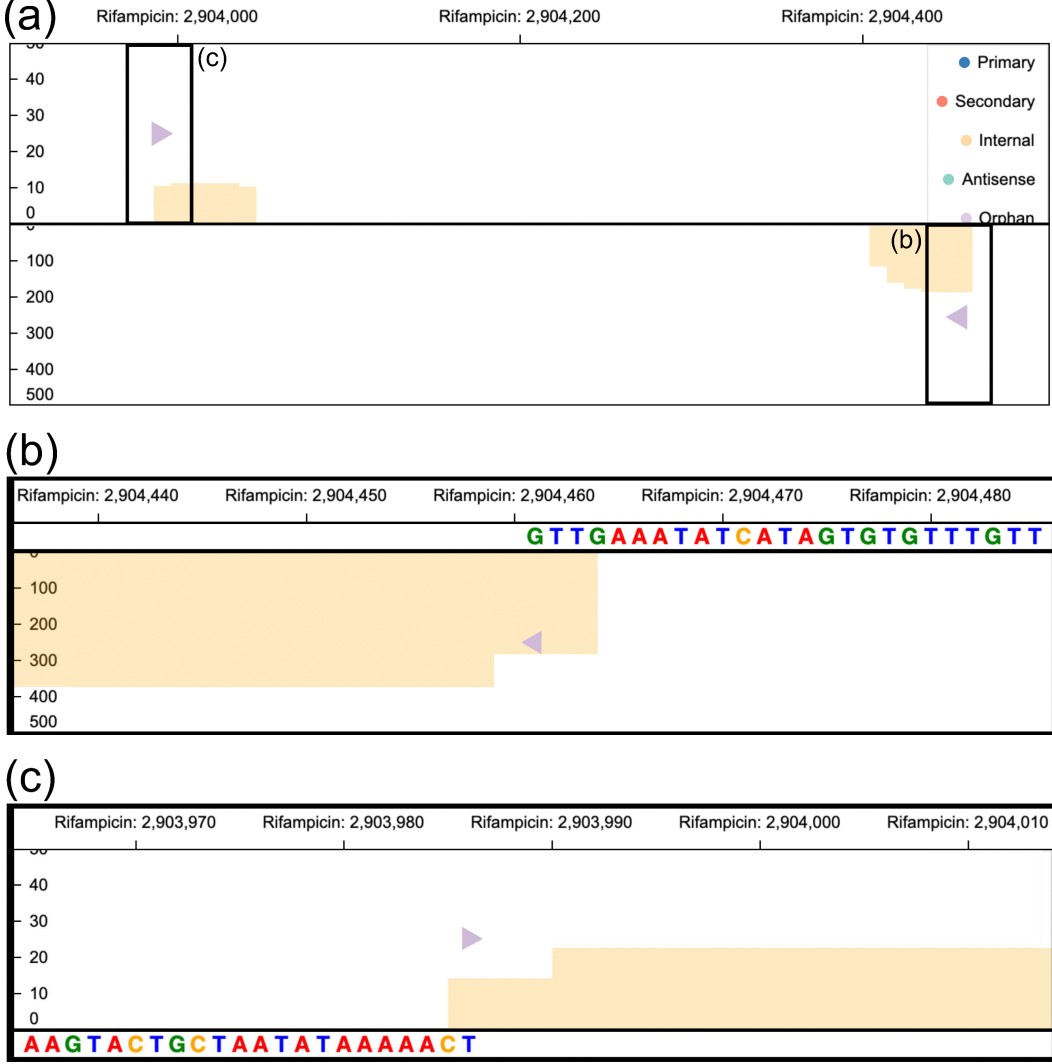

**Fig 8. Usage of the *genome browser* for the exploration and characterization of orphan TSS in *E. coli*.** For simplicity, empty visualization tracks have been removed from the figures. **(a)** *Genome viewer* on *Single view* mode visualizing a region with two orphan TSS that occur in all conditions, shown here only for the rifampicin treatment. On the reverse strand (bottom), the TSS position $2,904,461$ shows a prominent *step height* rising up to an expression of around 350. Interestingly, on the forward strand at position $2,903,986$ a further orphan TSS position was identified. **(b)** Zoomed view of TSS at position $2,904,461$ of the reverse strand with an expression value of the enriched library (orange bars) around 350. The upstream region shows a putative Pribnow box (`TATAAA`) starting at −9 nt upstream of the TSS. **(c)** Zoomed view of TSS at position $2,903,986$ with an expression value of the enriched library (orange bars) around 25. The upstream region shows again a clear putative Pribnow box (`TAATATAA`) starting at −13 nt upstream of the TSS.

expected from researchers with a biological background, especially since they are the individuals who generate insights from the data. Moreover, data exploration and integration are key steps in the generation of insights. However, this step has been neglected by many of the currently existing tools. Therefore, we defined requirements and goals to close the gap in insight generation for TSS prediction workflows at every step through the implementation of TSSPREDATOR-WEB.

As a web-application, the platform-independent usage and the user-friendly GUI of TSSPREDATOR-WEB enhance the accessibility of the prediction workflow (our defined goal **G1**). In addition, the web-based application allows for the

integration of modern, user-centered approaches that address the remaining goals defined for TSSPREDATOR-WEB. For example, it facilitates the reproducibility and data sharing of the workflow (**G5**) via the upload of a configuration file containing all the input files.

Besides reproducibility, TSSPREDATOR-WEB was developed to increase the usability of the TSS prediction workflow (**G2**). This enhanced usability begins with the improved and intuitive file upload process and continues with the asynchronous TSS prediction executed in the backend. A further major usability improvement is the ability to explore genomic and transcriptomic data directly within the interface. Users can now rely not only on TSSPREDATOR-WEB for TSS prediction but also for data exploration (**G3**), as illustrated in our use case. The exploration can be pursued from different angles. The *MasterTable* provides access to all predicted results of TSSPREDATOR, while the visualizations provide a more comprehensive view of the data. For example, a quick overview of the TSS distribution among classes, experiments, and locations in the genome can be achieved either by using the UpSet plot or by the aggregated view of the *genome browser*. As shown in the use case, users can identify specific TSS sets of interest; for example, those that occur only under specific conditions; such as those enriched only when *E. coli* was treated with antibiotics.

The aim of the *genome browser* is to provide a single view that integrates genomic and transcriptomic data (**G4**). Prior studies had to rely entirely on external tools, such as MEME, to provide a genomic context for a TSS. Though our implementation of TSSPREDATOR-WEB does not provide any statistical information on the occurrence of the sequences upstream of a TSS, it allows a general exploration and contextualization of the data. With this, users are not only able to identify TSS with high confidence values but also to dig deeper into their potential regulation, such as sequences present in their promoter regions.

Moreover, the ability to characterize orphan TSS, those TSS not linked to annotated transcriptional units, opens the door to discovering previously overlooked genomic elements, such as ncRNAs [43,44]. Using the genome browser of TSSPREDATOR-WEB, the confidence in such sites can be evaluated based on expression, high step height values, or the presence of a Pribnow box. Other tools, such as the recently developed pipeline TSS-CAPTUR [11], focus on further characterizing these TSS sites to provide a hint about the functionality of the transcript and to close the gap in missing gene annotations. Both TSSPREDATOR-WEB and TSS-CAPTUR are part of the Tübingen Visualization Server (TueVis) initiative, a visualization server for user-friendly tools. In the future, it is planned to directly link TSSPREDATOR-WEB with TSS-CAPTUR to allow a seamless transition from exploratory TSS prediction to detailed characterization of uncharacterized TSS.

While the current exploration within TSSPREDATOR-WEB is primarily visual, users may also be interested in identifying statistically different TSS between conditions. This can be achieved by exporting the normalized expression profiles generated by TSSPREDATOR and extracting the enriched coverage profile surrounding each enriched TSS position based on the *MasterTable*. These profiles are treated as distributions. Our statistical approach uses the Kolmogorov-Smirnov (KS-)test [45,46] by comparing the coverage distributions around the called TSS between the conditions. For this, we conduct pairwise KS-tests between all replicates of the conditions to be compared and compute a common $p$-value using the Cauchy combination test [47]. Finally, all the combined $p$-values of all tested TSS are subject to a Bonferroni $p$-value correction [48].

This approach has not yet been integrated into TSSPREDATOR-WEB, but we provide a preliminary implementation in the GitHub repository.

Moreover, TSSPREDATOR-WEB has so far been thoroughly tested using coverage profiles that account for the complete read length, particularly for defining the predefined parameter sets (see S1 Fig). However, *READemption* [19] and other coverage profile generating tools, such as `bedtools genomcov` [49] or `deeptools` [50], also allow for the possibility of accounting for only the first base of the read during profile generation. While TSSPREDATOR-WEB has not been fully tested on such coverage profiles, we expect it to correctly identify TSS in this data as well. Still, as first-base profiles tend to be less smooth, especially in intergenic regions, we recommend running the prediction with more specific thresholds for the *step height* and *step factor* parameters as the ones used for full-read coverage profiles.

Additionally, future developments of TSSPREDATOR-WEB may include support for newer sequencing protocols such as Cappable-ONT [42] for long-read-based TSS prediction and Term-seq [51] for the characterization of transcription termination sites (TTS). This would provide deeper insights into RNA processing, regulation and boundary detection, further extending the depth and applicability of TSS analyses in prokaryotes.

In conclusion, TSSPREDATOR-WEB provides an accessible and interactive platform for genome-wide TSS exploration, improving the discovery and interpretation of bacterial transcriptomics. Its user-friendly focus, combined with the capability of visual analysis of the data, provides users with a good basis for the analysis of the prokaryotic transcriptome architecture.

## Supporting information

**S1 Fig. Overview of the five predefined sets of thresholds for *Step height*, *Step factor* and *Enrichment factor* for the TSS prediction with TSSpredator.**
(TIF)

**S2 Fig. Overview of how different parameter sets influence TSS calling in TSSpredator.** Three discontinuous genomic positions (*i*, *j*, *k*) show loci with putative TSS. The top plot shows the normalized expression profiles of the enriched and non-enriched RNA-seq libraries for one condition and one replicate. The middle plots display the derived metrics *Step Height*, *Step Factor*, and *Enrichment Factor*, together with the corresponding thresholds for the *very specific*, *default*, and *very sensitive* parameter sets. Positions exceeding a given threshold are marked with a colored point according to that parameter set. Note that for *Step Factor*, the thresholds for the *very specific* and *default* sets coincide. In the bottom plot, a TSS is called at positions where all three metrics surpass the thresholds of a given parameter set. The color of each TSS indicates the parameter set that would identify it.
(TIF)

**S3 Fig. Screenshot of *drag & drop* process for file distribution.** Instead of selecting files individually, they can be uploaded at once and distributed across the different conditions and replicates.
(TIF)

**S4 Fig. Secondary structure of the UTR region of the gene *ugd* as predicted by RNAfold.**
(TIF)

**Supplementary Material. File containing the S1, S2 & S3 Algorithms.**
(PDF)

## Acknowledgments

We thank Valerie Bouillon for her contributions to the prototype of the user-interface for TSSPREDATOR-WEB. We also thank Dilek Tuncbilek-Dere and Sven Fillinger for further developments of TSSPREDATOR. Lastly, we thank Natalia Gogoleva and Yuri Gogolev for answering questions regarding the library preparation of the analyzed dataset.

## Author contributions

**Conceptualization:** Mathias Witte Paz, Alexander Herbig, Kay Nieselt.

**Data curation:** Mathias Witte Paz.

**Funding acquisition:** Kay Nieselt.

**Software:** Mathias Witte Paz, Alexander Herbig.

**Supervision:** Kay Nieselt.

**Visualization:** Mathias Witte Paz.

**Writing – original draft:** Mathias Witte Paz, Kay Nieselt.

**Writing – review & editing:** Mathias Witte Paz, Alexander Herbig, Kay Nieselt.

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
