## [Decision Letter · Decision Letter 0]

28 Oct 2025

Dear Dr. Nieselt,

Thank you for submitting your manuscript to PLOS ONE. After careful consideration, we feel that it has merit but does not fully meet PLOS ONE’s publication criteria as it currently stands. Therefore, we invite you to submit a revised version of the manuscript that addresses the points raised during the review process.

We look forward to receiving your revised manuscript.

Kind regards,

António Machado, PhD

Academic Editor

PLOS ONE

Journal Requirements:

“MWP and KN are supported by infrastructural funding from the Cluster of Excellence EXC 2124 ‘Controlling Microbes to Fight Infections’ [project ID 390838134] from the DFG (Deutsche Forschungsgemeinschaft, German Research Foundation).

We acknowledge support from the Open Access Publication Fund of the University of Tübingen.”

Please provide an amended statement that declares *all* the funding or sources of support (whether external or internal to your organization) received during this study, as detailed online in our guide for authors at http://journals.plos.org/plosone/s/submit-now . Please also include the statement “There was no additional external funding received for this study.” in your updated Funding Statement.

“MWP and KN are supported by infrastructural funding from the Cluster of Excellence EXC 2124 ‘Controlling Microbes to Fight Infections’ [project ID 390838134] from the DFG (Deutsche Forschungsgemeinschaft, German Research Foundation).

We acknowledge support from the Open Access Publication Fund of the University of Tübingen.”

5. We note that Figure 3 in your submission contain copyrighted image. All PLOS content is published under the Creative Commons Attribution License (CC BY 4.0), which means that the manuscript, images, and Supporting Information files will be freely available online, and any third party is permitted to access, download, copy, distribute, and use these materials in any way, even commercially, with proper attribution. For more information, see our copyright guidelines: http://journals.plos.org/plosone/s/licenses-and-copyright.

a. You may seek permission from the original copyright holder of Figure 3 to publish the content specifically under the CC BY 4.0 license.

6. We are unable to open your Supporting Information file S1_DragDrop.eps and S2_RNAstructure.eps. Please kindly revise as necessary and re-upload.

Additional Editor Comments:

All three reviewers recommeded only minor revisions. Congratulations! Please revised the manuscript accordingly to their comments.

Reviewers' comments:

Reviewer's Responses to Questions

**Comments to the Author**

1. Is the manuscript technically sound, and do the data support the conclusions?

Reviewer #1: Yes

Reviewer #2: Yes

Reviewer #3: Yes

2. Has the statistical analysis been performed appropriately and rigorously?

Reviewer #1: N/A

Reviewer #2: N/A

Reviewer #3: N/A

3. Have the authors made all data underlying the findings in their manuscript fully available?

Reviewer #1: Yes

Reviewer #2: Yes

Reviewer #3: Yes

4. Is the manuscript presented in an intelligible fashion and written in standard English?

Reviewer #1: Yes

Reviewer #2: Yes

Reviewer #3: Yes

Reviewer #1: TSSpredator is a useful tool for TSS mapping from prokaryotic RNA-seq data and this webportal is user-friendly in its design with a set of presets to alter sensitivity/specificity of the search. The replicate handling seems reasonable, and the user is provided with a range of files and visualisatio possibilities that make this webportal certainly valuable. The manuscript does a great job explaining all the options.

I cannot comment much on the building of the website as this is beyond my expertise.

There is one important point that remains unclear to me. The “coverage tracks” used by TSSpredator as input appear to be coverage tracks for the entire read length as judged from the figures 1, 4, 7 and 8. Limiting the coverage calculation to the 5’-ends of the reads that actually represent the TSS signal makes more sense and this option is included in the READemption software that the authors recommend for input file generation (“-b first_base_only”) as well as other widely used software like bedtools genomcov or deeptools . I know this rather refers to TSSpredator itself than the webportal described here. But, as a minimum, it should be properly defined in the text whether 5’-end coverage or read coverage (or fragment coverage for paired-end data?) is used.

Besides this point, I have only very minor comments.

Page 2, line 18 “most genome annotations only provide translation start sites” , maybe more precise to say “predicted translation start sites”, generally they are not experimentally verified.

Page 3, line 90 “pre-normalized” This is somewhat ambigous, raw coverage might be a better term here?

Page 8, line 286 Novobiocin is a DNA gyrase inhibitor, thus will affect more processes than “DNA synthesis”

Page 9, line 324 “subset of TSS: those enriched only in the presence of an antibiotic” In the absensce of any statistical testing whether there is actually a significant difference here, I would rather say “passing the TSS calling threshold”.

Reviewer #2: This manuscript is well articulated and provides clear guidance for any usage of this tool. One potential concern is that the algorithm underlying TSSPredator has been previously published, however the authors do not claim novelty with regard to this. In fact, the tool has been available for some time in a command-line interface. Rather what the manuscript offers is increased usability of this tool through a streamlined process and integration with data visualization resources, which was not previously available. Indeed the authors outline 5 goals for the development of this GUI and clearly address how they accomplished each one. Thus I think the manuscript potentially represents a valuable contribution to the community. Indeed the authors even informed their design of the tool based on common approaches used by other studies for data visualization, which is a thoughtful approach. However as the usability of the GUI is central to this project's impact it is important to note that the GUI is slow. This is not necessarily a problem for file upload and analysis as these are expected to take time, however when I was trying to work with the output the website kept crashing. This is a concerning issue for usability of this tool as one of its major assets is data visualization. As the user can already download a set of results, it is worth considering also producing scripts for figure generation to support data visualization that could be easily run locally such as in R Studio (see http://revigo.irb.hr/ for an example of this style of implementation). This could be a workaround of website bandwidth issues if other options are not possible.

Comments:

lines 17-19 consider addressing why standard RNA-seq does not produce comprehensive results for TSS prediction

lines 36-39 You make the claim that TSSs are essential for comparative transcriptomics. However comparative transcriptomics is accomplished without this. Consider more precise wording of what you mean by comparative transcriptomics.

Discussion - consider slightly polishing the wording here. While fine and readable it read less polished than the other sections of the manuscript.

Change file type (e.g., png, jpeg, tiff) for supplementary materials so it is easier to view and does not have to be separately rendered.

Reviewer #3: This manuscript presents the development of a user-friendly frontend for TSSPredator, a tool that identifies transcription starting sites from RNA sequencing data, called TSSPredator-Web. In addition to simplifying the prediction process for less technical researchers, TSSpredator-Web also provides an interface to explore the results dynamically. Finally, the authors demonstrate a use case of the tool to identify both known and potentially novel TSSs associated with antibiotic exposure.

This manuscript clearly describes the current state of the field and identifies the gap that their software is filling. The descriptions of the software and the TSSpredator algorithm are very clear. The use case showcases the capabilities of the software well and demonstrates clearly how a researcher could use it. However, the use case could be expanded a bit to compare TSSpredator with the other mentioned TSS identification approaches.

Overall I would recommend this manuscript be accepted with some minor revisions.

Though the authors describe the TSSpredator approach clearly in section 2.1, since it has not been formally published previously, they may consider making this work that publication and formalizing the algorithm, possibly using pseudo-code or taking a more mathematical approach. Additionally, Figure 1 could be slightly elaborated to include where certain parameter choices are implicated. In fact, the authors might consider integrating Figures 1 and 2 to provide a more complete view of the workflow.

The authors provide a use case of the tool in section 3 to demonstrate its capabilities over existing approaches. Near the end (line 355), the authors propose the potential identification of “previously unannotated transcriptional units.” However, it’s unclear if these are due to the new dataset or if they are unique to the TSSpredator algorithm. A comparison with the other aforementioned approaches would be useful. Overall, a summary of the differences between TSSpredator and the other approaches could put into context how the results from this approach differ.

The authors make clear that TSSpredator-Web is primarily for data exploration, and statistical evaluations of TSS identification are not elaborated on. However, potential researchers may be interested in demonstrating whether the existence of a particular TSS is statistically different between two or more experimental conditions. Though the tool may not be able to achieve this, it might be worth clarifying whether it would be possible to do this using the underlying data behind TSS identification. If not, it may be worth adding to the discussion of limitations.

**Do you want your identity to be public for this peer review?** For information about this choice, including consent withdrawal, please see our Privacy Policy

Reviewer #1: No

Reviewer #2: No

Reviewer #3: No

---

## [Author Response · Author response to Decision Letter 1]

1 Feb 2026

Journal Requirements

Reply: Our code is provided as a Github repository, with a DOI provided via Zenodo.

“MWP and KN are supported by infrastructural funding from the Cluster of Excellence EXC 2124 ‘Controlling Microbes to Fight Infections’ [project ID 390838134] from the DFG (Deutsche Forschungsgemeinschaft, German Research Foundation).

We acknowledge support from the Open Access Publication Fund of the University of Tübingen.”

Reply: We have adapted the funding and added the expected statement.

5. We note that Figure 3 in your submission contain copyrighted image. All PLOS content is published under the Creative Commons Attribution License (CC BY 4.0), which means that the manuscript, images, and Supporting Information files will be freely available online, and any third party is permitted to access, download, copy, distribute, and use these materials in any way, even commercially, with proper attribution. For more information, see our copyright guidelines: http://journals.plos.org/plosone/s/licenses-and-copyright.

a. You may seek permission from the original copyright holder of Figure 3 to publish the content specifically under the CC BY 4.0 license.

Reply: We are unsure to what extent this figure might contain copyrighted material, as we, ourselves, created this figure. It lists the requirements of TSSpredatorWeb and hence are unsure which other article should have published such a figure.

6. We are unable to open your Supporting Information file S1_DragDrop.eps and S2_RNAstructure.eps. Please kindly revise as necessary and re-upload.

Reply: we apologize for the inconvenience of using the postscript file format. We have now uploaded and thus provide all figures as PDFs.

Reply: We have revised the reference list and completed some missing DOIs. We also are not citing any retracted papers.

Additional Editor Comments:

All three reviewers recommended only minor revisions. Congratulations! Please revised the manuscript accordingly to their comments.

Reply: Thank you, we highly appreciate that.

5. Review Comments to the Author

Reviewer #1:

TSSpredator is a useful tool for TSS mapping from prokaryotic RNA-seq data and this webportal is user-friendly in its design with a set of presets to alter sensitivity/specificity of the search. The replicate handling seems reasonable, and the user is provided with a range of files and visualization possibilities that make this webportal certainly valuable. The manuscript does a great job explaining all the options.

I cannot comment much on the building of the website as this is beyond my expertise.

There is one important point that remains unclear to me. The “coverage tracks” used by TSSpredator as input appear to be coverage tracks for the entire read length as judged from the figures 1, 4, 7 and 8. Limiting the coverage calculation to the 5’-ends of the reads that actually represent the TSS signal makes more sense and this option is included in the READemption software that the authors recommend for input file generation (“-b first_base_only”) as well as other widely used software like bedtools genomcov or deeptools . I know this rather refers to TSSpredator itself than the webportal described here. But, as a minimum, it should be properly defined in the text whether 5’-end coverage or read coverage (or fragment coverage for paired-end data

Reply: We thank the reviewer for pointing this out. Indeed, READemption offers both options for the computation of the coverages. To the best of our knowledge, the parameter for the “first base only” has not been used in other scientific publications when analysing TSS enrichment data, in particular when using TSSpredator on the basis of the READemption software. So our knowledge so far TSSpredator has is mainly used with coverage profiles accounting for the full read coverage (the default option in READemption). We added this information now in the manuscript (line 98). However, we point out, and have added this also in the manuscript (lines: 477-481), since TSSpredator does not conduct the step of coverage profile computation itself, the user is free to upload coverage profiles which limits the coverage calculation to the 5’ends of the reads. Nevertheless, we also would like to mention that our internal tests revealed, when comparing both approaches, that such coverage profiles might be more noisy, and lead to many more TSS calls, at least when using the default parameters of TSSpredator. Users therefore would need to choose more specific parameters, which however is not a problem in our program. We have also added this aspect in the discussion of our manuscript (lines:481-484).

Besides this point, I have only very minor comments.

Page 2, line 18 “most genome annotations only provide translation start sites” , maybe more precise to say “predicted translation start sites”, generally they are not experimentally verified.

Reply: Indeed, thank you, we have adapted the wording (in the introduction line 18).

Page 3, line 90 “pre-normalized” This is somewhat ambigous, raw coverage might be a better term here?

Reply: We thank the reviewer for pointing this out. Indeed, the term “raw coverage” is not correct in this case. The raw coverage profile is one of the outputs that READemption provides after coverage computation. However, we use one of the other two coverage files that are normalized: either by the total number of aligned reads and multiplied by the lowest number of aligned reads of all considered libraries (coverage-tnoar_min_normalized), or by the total number of aligned reads and multiplied by one million (coverage-tnoar_mil_normalized). We consider this pre-normalized as in TSSpredator another normalization step happens. However, we understand that this term might not be clear enough and have therefore adapted the wording in the manuscript (lines 103-107).

Page 8, line 286 Novobiocin is a DNA gyrase inhibitor, thus will affect more processes than “DNA synthesis”

Reply: We thank the reviewer for pointing this out, stating that our statement was too general in this regard. Indeed, Novobiocin inhibits the GyrB subunit of the DNA gyrase and therefore affects - among other things processes - the DNA replication. We have adapted the statement for this antibiotic and also expanded on the effects of the other two antibiotics for a clearer differentiation of these 3 antibiotics (lines: 315-317).

Page 9, line 324 “subset of TSS: those enriched only in the presence of an antibiotic” In the absensce of any statistical testing whether there is actually a significant difference here, I would rather say “passing the TSS calling threshold”.

Reply: We agree with the reviewer that the word “enriched” could be misleading and be interpreted as the result of a statistical analysis. Hence, we have adapted the statement to state that these 43 TSS were identified based on our TSS calling strategy (lines 352-355).

Reviewer #2:

This manuscript is well articulated and provides clear guidance for any usage of this tool.

One potential concern is that the algorithm underlying TSSPredator has been previously published, however the authors do not claim novelty with regard to this. In fact, the tool has been available for some time in a command-line interface.

Rather what the manuscript offers is increased usability of this tool through a streamlined process and integration with data visualization resources, which was not previously available. Indeed the authors outline 5 goals for the development of this GUI and clearly address how they accomplished each one.

Thus I think the manuscript potentially represents a valuable contribution to the community. Indeed the authors even informed their design of the tool based on common approaches used by other studies for data visualization, which is a thoughtful approach. However as the usability of the GUI is central to this project's impact it is important to note that the GUI is slow. This is not necessarily a problem for file upload and analysis as these are expected to take time, however when I was trying to work with the output the website kept crashing. This is a concerning issue for usability of this tool as one of its major assets is data visualization.

As the user can already download a set of results, it is worth considering also producing scripts for figure generation to support data visualization that could be easily run locally such as in R Studio (see http://revigo.irb.hr/ for an example of this style of implementation). This could be a workaround of website bandwidth issues if other options are not possible.

Reply: We thank the reviewer, in particular also for thoroughly testing the platform. Indeed, unfortunately we also have noticed this problem while exploring the data within the genome explorer outside of university networks, where the upload bandwidth is usually significantly lower than the download bandwidth. This issue only appears when using the genome visualization, which makes use of the Gosling library, therefore, also unfortunately, there is not much margin for caching or optimization. However, we have reduced the amount of API calls for specific tracks and have also made the read coverage visualization optional, as this visualization requires the highest amount of resources. With this we hope that the usability is much enhanced. In addition, we would like to emphasize that TSSpredator is also available via Docker images that can be run locally and therefore there is no dependency on the internet connection and the application runs faster and more smoothly. We added that statement in the manuscript (lines 226-229)

Comments:

lines 17-19 consider addressing why standard RNA-seq does not produce comprehensive results for TSS prediction

Reply: Standard RNA-seq is insufficient for accurate TSS identification, since it cannot distinguish between primary and processed transcripts. When a substantial fraction of transcripts is processed, the resulting 5′-end signal appears as a ladder-like pattern, which makes the precise location of the true TSS more difficult and complicates reliable prediction. We have added this and thus expanded the text to explain this aspect in more detail in the manuscript (lines 18-25).

lines 36-39 You make the claim that TSSs are essential for comparative transcriptomics. However comparative transcriptomics is accomplished without this. Consider more precise wording of what you mean by comparative transcriptomics.

Reply: We thank the reviewer for this comment. Indeed, this wording was not ideal. We have adapted our statement for a more precise formulation on how TSS can be applied for comparative transcriptomics (lines 40-44).

Discussion - consider slightly polishing the wording here. While fine and readable it read less polished than the other sections of the manuscript.

Reply: We have polished the discussion to improve readability (hopefully).

Change file type (e.g., png, jpeg, tiff) for supplementary materials so it is easier to view and does not have to be separately rendered.

Reply: Indeed, and we apologize for the inconvenience of using postscript as a file format. We have adapted the figures to a more compatible format (PNG & PDF).

Reviewer #3:

This manuscript presents the development of a user-friendly frontend for TSSPredator, a tool that identifies transcription starting sites from RNA sequencing data, called TSSPredator-Web. In addition to simplifying the prediction process for less technical researchers, TSSpredator-Web also provides an interface to explore the results dynamically. Finally, the authors demonstrate a use case of the tool to identify both known and potentially novel TSSs associated with antibiotic exposure.

This manuscript clearly describes the current state of the field and identifies the gap that their software is filling. The descriptions of the software and the TSSpredator algorithm are very clear. The use case showcases the capabilities of the software well and demonstrates clearly how a researcher could use it.

Reply: We thank the reviewer for this positive comment.

However, the use case could be expanded a bit to compare TSSpredator with the other mentioned TSS identification approaches. Overall I would recommend this manuscript be accepted with some minor revisions.

Though the authors describe the TSSpredator approach clearly in section 2.1, since it has not been formally published previously, they may consider making this work that publication and formalizing the algorithm, possibly using pseudo-code or taking a more mathematical approach.

Reply: We thank the reviewer for their suggestion. We have now expanded the explanation of the algorithm and have also included a pseudocode of the key steps (see Suppl. Algorithms 1-3).

Additionally, Figure 1 could be slightly elaborated to include where certain parameter choices are implicated. In fact, the authors might consider integrating Figures 1 and 2 to provide a more complete view of the workflow.

Reply: We thank the reviewer for this suggestion. Since both Figures 1 and 2 do not really address the question of how

---

## [Decision Letter · Decision Letter 1]

16 Feb 2026

TSSpredator-Web: A web-application for transcription start site prediction and exploration

PONE-D-25-28983R1

Dear authors,

I am pleased to inform you that the revised manuscript was accepted for publication by all three reviewers.

Thank you for submitting your work to the PLOS ONE journal and best regards,

António Machado

Reviewers' comments:

Reviewer's Responses to Questions

**Comments to the Author**

Reviewer #1: All comments have been addressed

Reviewer #2: All comments have been addressed

Reviewer #3: All comments have been addressed

2. Is the manuscript technically sound, and do the data support the conclusions?

Reviewer #1: Yes

Reviewer #2: Yes

Reviewer #3: Yes

3. Has the statistical analysis been performed appropriately and rigorously?

Reviewer #1: Yes

Reviewer #2: N/A

Reviewer #3: Yes

4. Have the authors made all data underlying the findings in their manuscript fully available?

Reviewer #1: Yes

Reviewer #2: Yes

Reviewer #3: Yes

5. Is the manuscript presented in an intelligible fashion and written in standard English?

Reviewer #1: Yes

Reviewer #2: Yes

Reviewer #3: Yes

Reviewer #1: Congratulations to the authors, I just want to one comment the authors might take into consideration:

p.2, l.19: Besides the inability of standard RNA-seq to distinguish processed 5' ends, it might be worth mentioning the 5' end fragmentation problem resulting from RNA fragmentation prior to reverse transcription and leading generally to lower 5' end coverage compared to the middle body of the gene.

Reviewer #2: The authors have adequately and thoughtfully address my comments as well as those of the other reviewers.

Reviewer #3: (No Response)

**Do you want your identity to be public for this peer review?** For information about this choice, including consent withdrawal, please see our Privacy Policy

Reviewer #1: **Yes:** Fabian Blombach

Reviewer #2: No

Reviewer #3: No

---

## [Editor Report · Acceptance letter]

PONE-D-25-28983R1

PLOS One

Dear Dr. Nieselt,

I'm pleased to inform you that your manuscript has been deemed suitable for publication in PLOS One. Congratulations! Your manuscript is now being handed over to our production team.

Kind regards,

on behalf of

Dr. António Machado

Academic Editor

PLOS One